

# No population bias to left-hemisphere language in 4-year-olds with language impairment

Dorothy V.M. Bishop[1], Georgina Holt[1], Andrew J.O. Whitehouse[1,2] and Margriet Groen[1,3]

[1] Department of Experimental Psychology, University of Oxford, UK
[2] Telethon Institute of Child Health Research, University of Western Australia, Perth, WA, Australia
[3] Radboud University, ED Nijmegen, Netherlands

Corresponding author
Dorothy V.M. Bishop,
dorothy.bishop@psy.ox.ac.uk

## ABSTRACT

**Background.** An apparent paradox in the field of neuropsychology is that people with atypical cerebral lateralization do not appear to suffer any cognitive disadvantage, yet atypical cerebral lateralization is more common in children and adults with developmental language disorders. This study was designed to explore possible reasons for this puzzling pattern of results.

**Methods.** We used functional transcranial Doppler ultrasound (fTCD) to assess cerebral blood flow during language production in 57 four-year-olds, including 15 children who had been late-talkers when first seen at 20 months of age. We categorized cerebral lateralization as left, right or bilateral, and compared proportions with each type of laterality with those seen in a previously tested sample of children aged 6–16 years. We also compared language scores at 4 years for those with typical and atypical lateralization, and then looked at the association the opposite way: comparing those with typical or impaired language in terms of their cerebral lateralization.

**Results.** The distribution of types of cerebral lateralization was similar for 4-year-olds to that seen in older children. Overall, cerebral lateralization was not predictive of language level. However, for children who had language difficulties at 20 months and/or 4 years ($N = 21$), there was no population bias to left-hemisphere language activation, whereas children without language problems at either age showed a pronounced bias to left-sided language lateralization. Nevertheless, many children with right hemisphere language had no indications of language difficulties, confirming that atypical cerebral asymmetry is not a direct cause of problems.

**Conclusions.** We suggest that atypical lateralization at the individual level is not associated with language impairment. However, lack of lateralization at the population level is a marker of risk for language impairment, which could be due to genetic or non-genetic causes.

## BACKGROUND

It is well established that language processing in the brain is asymmetric, with the left cerebral hemisphere playing the major role in most people. However, a minority of people

do not follow the usual pattern, and either have right-hemisphere language, or no bias toward either hemisphere. This functional asymmetry is not readily predictable from structural asymmetry of the two hemispheres (*Dorsaint-Pierre et al., 2006*; *Keller et al., 2011*; *Propper et al., 2010*), and its significance for language development remains poorly understood (*Bishop, 2013*). It is often assumed that a lateralized brain evolved because it conferred some advantage on the individual, perhaps by enabling division of labour between the hemispheres (*Vallortigara, 2006*), yet in humans, lateralization is not universal.

Over seventy years ago, *Orton (1937)* proposed that a failure to develop the usual left-hemisphere language bias could be detrimental for language development; although there have been numerous studies on this topic since that time, most relied on handedness as an indicator of cerebral lateralization, and results have been contradictory and confusing (*Bishop, 1990*; *Bishop, 2005*). Handedness, however, is an indirect and imprecise indicator of cerebral lateralization for language, regardless of whether it is measured by preference or performance measures (*Groen et al., 2013*).

We are now in a position to use more direct assessment of language laterality, but to date there has been little evidence of disadvantages associated with atypical lateralization in non-clinical samples (*Berl et al., 2014*; *Knecht et al., 2001*). Nevertheless, reduced bias to left-hemisphere language has been reported in several functional imaging studies of developmental language delay or disorder (*Badcock et al., 2011*; *Bernal & Altman, 2003*; *Chiron et al., 1999*; *Dawson et al., 1989*; *de Guibert et al., 2011*; *Ors et al., 2005*; *Tzourio et al., 1994*; *Whitehouse & Bishop, 2008*) and in young children with weak reading skills (*Bach et al., 2010*). However, a concern is that this result could be artefactual if degree of observed lateralization depends on task performance. For instance, lateralization could be affected by the amount of effort required when doing a language task (*Berl, Vaidya & Gaillard, 2006*; *Kadis et al., 2011*); alternatively, the amount of engagement of left hemisphere systems might depend on the amount of language generated (*Badcock, Nye & Bishop, 2012*). If so, we might expect to see a reduction in left-hemisphere lateralization in those with less well-developed language skills, just because they are poor at doing the task used for language activation.

The question of whether cerebral lateralization is related to language impairment is complicated by developmental issues. *Lenneberg (1967)* proposed that early in life both hemispheres participate in language functions, with cerebral lateralization developing gradually as language is acquired. He argued that this could explain why children make better recovery than adults from aphasia caused by focal left-brain injury; the idea is that the right hemisphere continues to play a role in language processing up to puberty, and so is more readily able to take over language functions after brain damage. According to this view, lack of language lateralization could be viewed as a form of neurodevelopmental immaturity. This notion has, however, been challenged by subsequent studies showing that left-sided language processing is usually evident from early childhood, leading *Witelson (1987)* to conclude that it is not cerebral lateralization that increases with age but "the amount of cognition available to be asymmetrically mediated by the hemispheres" (p. 679). More recent studies using functional imaging have generally supported the notion

that the left hemisphere usually mediates language production from early childhood, but there is still debate as to whether the extent of right hemisphere involvement declines with age. Most studies have used relatively small cross-sectional samples which lack power to detect more subtle changes in extent of lateralization. In a large cross-sectional study using fMRI, a small but significant increase in left-sided lateralization for verb generation was observed in Broca's area from age 5 years to adulthood ($r = .311$) (*Holland et al., 2007*). An fMRI study using a covert verb generation task with over 300 children aged 5–18 years found that task-related activation could be subdivided into seven independent components, only one of which was strongly lateralized (*Karunanayaka et al., 2010*). This lateralized component, which involved medial temporal gyrus, frontal gyrus, inferior frontal gyrus and angular gyrus, showed age-related increases in activation ($r = .42$). However, this result seems task-dependent; when a similar analysis was done using a word-picture matching task, a strongly lateralized component again emerged, with similar topography, but this did not show any correlation with age (*Schmithorst, Holland & Plante, 2007*). Other, smaller studies have noted developmental changes in lateralization of neuromagnetic (*Kadis et al., 2011*; *Ressel et al., 2008*) or BOLD responses (*Berl et al., 2014*) over childhood, although none to date has demonstrated this longitudinally, and there are also failures to find age trends (*Gaillard et al., 2000*; *Wood et al., 2004*). Ideally, longitudinal data are needed to study developmental trends. *Szaflarski et al. (2006b)* studied 30 children seen annually over a five-year period using fMRI with a verb generation task. There was an increase in activation of left inferior frontal gyrus between 5 and 12 years of age, which the authors interpreted as indicating increased lateralization with age (*Szaflarski et al., 2006b*). However, other left-sided regions showed decreased activation with age. Ideally, one would want to see an analysis of change in a laterality index with development (*Nieuwenhuis, Forstmann & Wagenmakers, 2011*). A further report from this sample measured fMRI using a narrative comprehension task over a 10-year period, but this was not informative about age-trends in lateralization because activation (in superior temporal lobes) was consistently bilateral (*Szaflarski et al., 2012*).

One reason why there are few large-scale studies of development of cerebral lateralization is because for many years the only way to assess cerebral lateralization in nonclinical samples was by functional brain imaging, which is not commonly undertaken with those under 7 or 8 years of age. An alternative procedure, functional transcranial Doppler ultrasound (fTCD), offers a cost-effective alternative that gives similar results to fMRI, when the same activation task is used (*Deppe et al., 2000*). Lohmann and colleagues (*2005*) pioneered the use of fTCD with children, and demonstrated it was possible to obtain a reliable index of language laterality in some children as young as two years of age. Their method required children to sit still and quiet with their eyes closed for a 30 s interval between stimuli, a protocol that can be taxing for young and distractible children. We developed a less demanding task which involves the child silently watching a language-free video clip during a baseline period, and then describing what had happened (*Bishop, Watt & Papadatou-Pastou, 2009*). This procedure is repeated for a maximum of 30 trials, and changes in cerebral circulation while the child is speaking are monitored via Doppler

ultrasound probes positioned to record blood flow in the left and right middle cerebral arteries.

Here we report data from a group of 57 children who were recruited for a study of language development when they were 20 months of age (*Bishop et al., 2012*), with late-talkers being oversampled. When they were 4 years old, these children were reassessed and classified according to whether or not they had language impairments. FTCD was also used at this age to assess cerebral lateralization, and the proportion of children with left-biased language processing was compared with children at later ages.

The current study focuses on three questions.

1. Does cerebral lateralization become more established as language develops, i.e., is left-sided cerebral lateralization as reliable and frequent in 4-year-olds as it is in older children? To address this question we compared the 36 4-year-olds with no evidence of language difficulties with 51 children aged 6–16 years who were studied by *Groen et al. (2012)*.

2. If we subdivide 4-year-olds according to whether or not they have left-sided language lateralization, do these subgroups differ in language ability? We noted above that prior studies have failed to show such an association in adults. If normal language is seen in 4-year-olds with atypical cerebral lateralization, this suggests it is not a direct cause of children's language impairments.

3. Is cerebral asymmetry reduced in children with language impairment at 4 years of age? Although this is similar to question 2, previous research suggests a different answer may be obtained when the association is studied from this direction. If we replicate this association in children as young as 4 years, we need then to consider which causal models might account for it.

## MATERIALS AND METHODS

### Participants

#### *4-year-old children*

Figure 1 shows the composition of the sample, which is described in detail by *Bishop et al. (2012)*. Mothers were recruited from a maternity ward, excluding any whose babies had health problems. Children were first identified at 18–19 months of age on the basis of parental responses to a British adaptation of the MacArthur–Bates Communicative Development Inventory (CDI) (*Hamilton, Plunkett & Schafer, 2000*). Over a 15 month period, we recruited all available children ($N = 26$) who met criteria as late talkers, defined as having an expressive CDI more than one SD below the mean (10 words or less), and 70 average talkers, scoring between the 20th and 75th centile (range 13–196 words). Twenty-four of the late talkers and 58 of the average talkers were available for follow-up at 4 years of age. Parental consent for fTCD was given for 19 of the late-talkers and 47 of the average talkers, and useable data from this procedure (see below) was obtained for 15 late-talkers and 42 average talkers (a success rate of 86%).

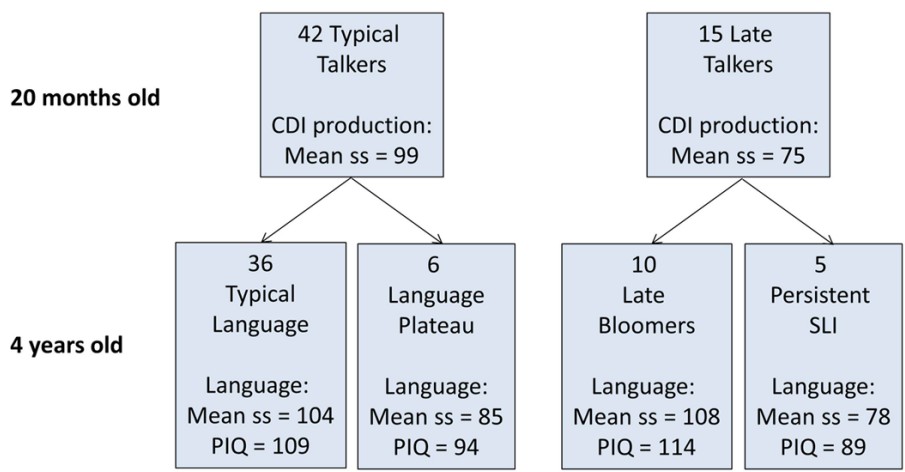

**Figure 1 Flowchart showing categorization of 57 children at 20 months and 4 years of age.** CDI production is scaled score corresponding to number of different words produced at 20 months, assessed using the McArthur Communicative Development Index. Language is scaled score corresponding to a principal component from a battery of six language tests given at 4 years of age (see Methods).

At follow-up at 4 years of age, children were categorized as having typical or impaired language on the basis of performance on a battery of tests that yielded nine language measures: Verbal Comprehension and Naming from the British Ability Scales (BAS) (*Elliott, Smith & McCulloch, 1997*), Sentence Repetition and Nonword Repetition from the Grammar and Phonology Screening Test (GAPS) (*Gardner et al., 2006*), Information and Sentence Length indices from the Bus Story test (*Renfrew, 1991*), third person singular and past tense measures from the Test of Early Grammatical Impairment (TEGI) (*Rice & Wexler, 2001*), and General Communication Composite from the Children's Communication Checklist-2 (*Bishop, 2003*). A principal component was extracted from the first six of these language measures to provide the summary score of language ability shown in Fig. 1, which was the factor score of the first component scaled to mean of 100 and SD of 15. For categorizing children's outcomes, language impairment was identified when the child scored more than 1 SD below the normative mean on two or more of the language measures. In addition, the Block Design and Matrices subtests of the Wechsler Preschool and Primary Scales of Intelligence, 3rd UK edition (WPPSI-III) (*Wechsler, 2002*) were used to give a prorated estimate of performance (nonverbal) IQ (PIQ).

### *School-aged children*

Full details of this sample are provided by *Groen et al. (2012)*. Participants were 34 boys and 28 girls spanning three age bands 6–8 years, 10–11 years, and 13–16 years of age recruited from schools around Oxfordshire, UK. Useable data from fTCD were obtained from 51 children (82%). These children completed a battery of verbal and nonverbal ability tests that confirmed they were functioning within the normal range.

In both samples children were excluded if they were affected by any known neurological disorder, had a diagnosis of autism or dyslexia, or did not have English as their primary home language.

### Ethical aspects

This project was approved by Oxfordshire Research Ethics Committee A, file number A03.025. Parents gave signed consent for participation by their child.

## Procedure

All participants completed two testing sessions, with behavioural testing of language and nonverbal ability in the first session, and fTCD in the second session. For 4-year-olds the ultrasound session took place in a quiet room at the University. Older children were tested in a quiet room at home or school or in a testing van.

### Handedness assessment

In both samples two types of handedness assessment were used, as described by *Groen et al. (2013)*. The first was based on the Edinburgh Handedness Inventory (EHI) (*Oldfield, 1971*). Children were asked to indicate the hand used for nine activities: writing, drawing, throwing, using a toothbrush, using a knife (without a fork), using a spoon, holding a broom (upper hand), opening a box (hand used to hold the lid) and dealing cards. Responses ($L$ or $R$) were converted to a handedness quotient $100 * (R - L)/(R + L)$, with positive numbers indicating right-handedness, and negative numbers left-handedness. The second was the Quantification of Hand Preference (QHP) task, which provides a behavioural measure of persistence of hand preference across the midline (*Bishop et al., 1996*). In this task, stacks of three cards with brightly coloured pictures are placed in seven spatial locations (approximately 30 degrees apart) along a semi-circle, on a table, within the child's reach. The child was located in the center of the semi-circle and asked to pick up a specific card and place it in a box located directly in front of them, without time constraints. The 4-year-olds were standing, whereas older children were seated in front of the table. The card order was random, but the sequence of positions was the same for all participants. The child was not informed of the experimenter's interest in hand preference, and treated the task as one of finding the named picture. The dependent variable was a laterality quotient (LQ), calculated by subtracting 0.50 from the proportion of right-hand reaches. This score ranged from $+0.50$ for participants reaching exclusively with the right hand through 0 for children who did not show a preference to $-0.50$ for those reaching exclusively with the left.

### Functional transcranial Doppler ultrasound (fTCD)

Blood flow velocity through the right and left middle cerebral arteries was measured with a Doppler ultrasonography device (DWL Multidop T2; DWL Elektronische Systeme, Singen, Germany). Children were fitted with a flexible head-set, which held in place two 2-MHz transducer probes, one over each temporal skull window. The experimental paradigm was controlled by Presentation Software (Neurobehavioral Systems) on a Dell laptop computer, which sent markers to the Multidop system to denote the start of each epoch.

The animation description paradigm is described in detail by *Bishop, Badcock & Holt (2010)*. Children watched clips from a specially designed cartoon which included sounds but no speech. Each trial started with the 12 s cartoon which the participant was asked to

watch silently. This was used as a baseline period: we previously established that there is no evidence of lateralized activation while participants passively watch these animations. Next, a response cue indicated the start of a 10 s period during which the screen showed a question mark, and the child described the part of the story shown in the preceding clip. This was followed by an 8 s silent rest period during which the screen showed a picture of a child with a finger to his lips. A maximum of 30 clips was used; we aimed to complete at least 20 trials with each child, but did more if the child was happy to continue and there was no time pressure. The child's verbal responses were audio-recorded. Because 4-year-olds did not always talk during the 'talk' period, we transcribed the session for these children and noted for exclusion any trials where the child either talked during the baseline period, or was silent during the activation period. In addition, for these children we computed the number of utterances and mean length of utterance in words during the 'talk' phases of valid trials.

### Analysis of data from fTCD

All data gathered with this paradigm were analyzed using the same Matlab program: this meant that the data from the older children were reanalyzed to ensure comparable procedures were followed. The principal difference from prior analysis was the introduction of an initial screening step to ensure that trials with any signal drop-out or spiking were either excluded, or (if the dropout/spiking affected only one data point) corrected by substituting the mean amplitude for that channel. Dropout was defined for each epoch as signal amplitude that fell to zero or more than 3 SD below the mean level for that channel, and spiking as signal amplitude more than 4 SD above the mean level for the channel. This process was followed after downsampling the signal to 25 Hz, and identifying epochs in relation to signal markers, but prior to other processing. The remaining processing steps followed methods previously described (*Badcock et al., 2012*; *Deppe, Ringelstein & Knecht, 2004*). These consist of normalization of both channels to a mean of 100, application of a heart cycle integration algorithm to smooth the phasic activity associated with the heartbeat, exclusion of epochs with extreme values, standardizing both channels in relation to the baseline period, and averaging across all accepted epochs. Note that, unlike in fMRI, a control task is not used when assessing laterality with fTCD. Rather, a period of inactivity is used to produce a stable baseline with similar levels of blood flow on left and right, and subsequently, in the data processing, signals from both sides are baseline-corrected to this period, so at the point when the language stimulus is presented, the two sides are equalized. The two values used in computation of a laterality index are the signals from left and right probes during an activation phase, baseline-corrected relative to the baseline period.

On the basis of previous research (*Groen et al., 2012*), we required a minimum of 12 accepted trials in the fTCD procedure as the basis of a laterality index. The laterality index (LI) is calculated as the mean blood flow velocity difference in a 2 s window centred on the peak difference value during a period of interest. The period of interest was 4–14 s after the onset of the cue to speak, as used previously with this paradigm (*Bishop, Watt & Papadatou-Pastou, 2009*). A positive LI indicates greater left than right hemisphere activation, and a negative index indicates predominantly right hemisphere lateralization.

The LI is computed separately for each epoch, making it possible to compute the standard error of the mean LI, and hence to determine whether an individual's LI is significantly different from zero, allowing for a categorical classification of language lateralization as left, right or bilateral. Bilateral language was identified if the 95% confidence interval for the LI spanned zero. In addition, the average LI was computed separately for odd and even trials, using the peak latency from the overall mean LI. This allows one to calculate an index of reliability of the LI in the sample as a whole, based on the intraclass correlation for the LI from odd and even trials.

## RESULTS

### Developmental trends in language lateralization in typically-developing children

For the analysis of age trends in cerebral lateralization, the 36 children with typical language development at both 20 months and 4 years of age were compared with the 51 older children studied by *Groen et al. (2012)*. As is evident from Table 1, the rates of left-hemisphere language and right-hemisphere language were not statistically different in the four age bands. Furthermore, the measurement of lateralization appeared adequately reliable in younger children; the intraclass correlation coefficients were high in all groups, with overlapping confidence intervals. Also, unreliable measurement would lead to more cases being classified with bilateral language; in fact, rates of bilateral language were no higher in the 4-year-olds than in older groups. Overall these results extend the previous finding of *Groen et al. (2012)*, who found no developmental trend in language-related lateralization of blood flow from 6 to 12 years.

### Comparison of language ability in those with and without left-sided language lateralization

This analysis focused just on the 4-year-old children, who were divided into two groups (regardless of language status): those with left-hemisphere language versus those with bilateral or right-hemisphere language (atypical laterality). Mean scores for these groups were compared using t-tests for standard scores from assessments at 20 months and 4 years. In no case did the difference in means approach statistical significance (see Table 2). A scatterplot showing the relationship between CDI word production scores at 18–19 months and laterality index at 4 years is provided in Supplemental Information 1.

### Language lateralization in 4-year-olds in relation to current language status

This analysis again focused just on 4-year-olds, but this time grouped children according to language status. Children with language difficulties at 4 years of age (i.e., the language plateau group and those with persistent SLI) were grouped together for comparison with the 4-year-olds with no language difficulties (those with typical language and late bloomers). Results are shown in Table 3. There was a significant difference in the proportions with left-, bilateral and right-hemisphere language in these two groups, though on the quantitative laterality index (LI), the difference between the two groups

**Table 1  Language laterality measures from functional transcranial Doppler ultrasound and handedness indices compared for four age groups.**

|  | 4 yr | 6–8 yr | 10–11 yr | 13–16 yr | Statistic | p |
|---|---|---|---|---|---|---|
| N | 36 | 20 | 17 | 14 |  |  |
| Mean (SD) age in yr | 4.07 (0.05) | 6.98 (0.47) | 10.82 (0.44) | 14.04 (0.82) |  |  |
| **Functional transcranial Doppler ultrasound** |  |  |  |  |  |  |
| N (%) left-lateralized | 26 (72%) | 12 (60%) | 12 (60%) | 9 (64%) | $\chi^2 = 2.2, df = 6$ | .906 |
| N (%) bilateral | 6 (17%) | 3 (15%) | 2 (12%) | 2 (14%) |  |  |
| N (%) right-lateralized | 4 (11%) | 5 (25%) | 3 (18%) | 3 (21%) |  |  |
| *Mean (SD) values* |  |  |  |  |  |  |
| N trials completed | 24.6 (4.98) | 18.3 (2.83) | 18.2 (2.43) | 19.2 (1.97) | $F(3, 83) = 19.05$ | <.001 |
| Laterality index (LI) | 2.76 (3.75) | 1.61 (3.50) | 2.58 (3.77) | 1.81 (2.86) | $F(3, 83) = 0.57$ | .635 |
| Peak latency | 9.2 (2.89) | 8.4 (2.46) | 9.0 (2.98) | 8.7 (2.38) | $F(3, 83) = 0.472$ | .703 |
| Odd/even reliability of LI | .906 | .963 | .933 | .944 |  |  |
| (ICC with 95% CI) | [.817–.952] | [.897–.986] | [.816–.976] | [.809–.982] |  |  |
| **Handedness** |  |  |  |  |  |  |
| Quantification of hand preference | 0.28 (0.15) | 0.24 (0.18) | 0.29 (0.26) | 0.22 (0.28) | $F(3, 83) = 1.346$ | .265 |
| Handedness inventory | 73.6 (30.83) | 54.0 (36.96) | 70.2 (45.88) | 56.0 (57.32) | $F(3, 83) = 0.401$ | .752 |

**Table 2  Mean (SD) scores from assessments at 20 months and 4 years in relation to language laterality.** On *t*-test none of the differences between means is significant at .05 level.

|  | Language laterality | |
|---|---|---|
|  | Left (N = 32) | Non-left (N = 25) |
| **20-month-old measures** |  |  |
| CDI word production, z score | −0.42 (0.758) | −0.63 (0.867) |
| CDI, word comprehension, z score | −0.30 (1.086) | −0.23 (0.895) |
| Mullen Receptive Language, T-score | 58.3 (10.33) | 58.5 (11.58) |
| Mullen Expressive Language, T-score | 49.0 (9.67) | 43.7 (10.82) |
| Vineland Communication scaled score | 98.8 (8.30) | 98.9 (9.91) |
| **4-year-old measures** |  |  |
| WPPSI short form PIQ | 106.3 (15.35) | 106.6 (17.69) |
| British Ability Scales, comprehension z-score | 0.39 (0.926) | 0.51 (1.051) |
| British Ability Scales, naming z-score | 1.21 (0.826) | 1.08 (0.804) |
| GAPS sentence repetition, z-score | 0.67 (0.965) | 0.80 (1.125) |
| GAPS nonword repetition, z-score | 0.64 (1.035) | 0.24 (1.049) |
| Bus Story information, scaled score | 97.4 (18.03) | 98.0 (20.50) |
| Bus Story sentence length, scaled score | 107.4 (19.93) | 107.1 (19.97) |
| TEGI, 3rd person singular, % produced | 0.91 (0.172) | 0.80 (0.304) |
| TEGI, past tense -ed, % produced | 0.90 (0.166) | 0.87 (0.226) |
| General Communication Composite CCC-2, z-score | 0.09 (0.934) | 0.04 (0.833) |

**Table 3 Measures from functional transcranial Doppler ultrasound and handedness indices in relation to language status at 4 years.**

| | Typical language | Language impaired | Statistic | $p$ |
|---|---|---|---|---|
| *N* | 46 | 11 | | |
| **Functional transcranial Doppler ultrasound** | | | | |
| *N* (%) left-lateralized | 29 (63.0%) | 3 (27.3%) | $\chi^2 = 4.03, df = 1$ | .045 |
| *N* (%) bilateral | 9 (19.6%) | 4 (36.4%) | (linear-by-linear) | |
| *N* (%) right-lateralized | 8 (17.4%) | 4 (36.4%) | | |
| *Mean (SD) values* | | | | |
| Laterality index[a] | 2.2 (3.95) | −0.2 (3.31) | $W = 333$ | .106 |
| *N* trials completed | 23.4 (5.69) | 21.6 (4.41) | $F(1, 55) = 0.98$ | .327 |
| Peak latency | 9.1 (2.97) | 10.8 (3.19) | $F(1, 55) = 2.92$ | .093 |
| Odd/even reliability | .912 | .950 | | |
| (ICC with 95% CI) | [.840–.951] | [.814–.987] | | |
| *N* utterances[a] | 27.6 (15.24) | 20.2 (11.93) | $F(1, 55) = 2.26$ | .138 |
| Mean length of utterance in words[b] | 5.11 (0.80) | 4.18 (1.27) | $F(1, 55) = 9.26$ | .004 |
| **Hand preference** | | | | |
| Quantification of hand preference[c] | 0.27 (0.153) | 0.24 (0.207) | $F(1, 53) = 0.24$ | .626 |
| Handedness inventory | 67.7 (45.57) | 46.2 (50.38) | $F(1, 55) = 1.89$ | .174 |

**Notes.**
[a] Mann–Whitney test used because data bimodal.
[b] Based on story-telling task during the transcranial Doppler ultrasound procedure.
[c] Missing data for two cases.

fell short of significance. For those with no language difficulties at 4 years, there was a significant bias to left-sided language, $t(45) = 3.80, p < .001$, whereas for those with language difficulties at 4 years, the mean laterality index did not differ significantly from zero, $t(10) = -0.20, p = .850$.

Before accepting these results at face value, it is necessary to consider possible confounding factors that might lead to a spurious association. First, we asked whether children with current language difficulties might show weaker laterality because they completed fewer trials and so had less reliable data. This did not seem the case. Although the children with language difficulties completed fewer trials on average on the fTCD procedure, this could not explain their reduced asymmetry, as the number of trials did not correlate significantly with the LI, Pearson $r = .105, p = .438$. Furthermore, the split-half reliability of the LI, calculated from the intraclass correlation between odd and even trials, was high and closely similar for the two groups. Also, the trend for weaker right-handedness in the language-difficulties group on the handedness inventory could not account for the LI difference, since this variable did not correlate significantly with LI either, Spearman's rho $= .196, N = 57, p = .143$. Overall, the majority of children in both groups were right-handed for writing: 39 of 46 (85%) of those with no language difficulties and 9 of 11 (90%) of those with language difficulties, a non-significant

difference, $\chi^2 = 0.3, d.f. = 1, p = .859$. (One case had missing data.) Thus it does not seem possible to explain the high rate of atypical language laterality in the language-difficulties group either in terms of less reliable measurement, or because of adventitious inclusion of non-right-handers.

An important question is whether reduced left-hemisphere lateralization is seen because the group with language difficulties produced fewer words during the fTCD activation procedure. The difference between groups was significant for mean length of utterance, but neither this variable, nor the number of utterances, correlated significantly with the LI; Pearson $r = 11, p = .399$ for number of utterances, and $r = -.03, p = .852$ for mean length of utterance. Note too that we excluded trials where the child said nothing during the activation procedure.

Because of the small sample size, we had grouped together all children with language difficulties at 4 years, regardless of their status at 20 months. It is of interest, nevertheless, to explore the data further to see whether reduced lateralization is a particular characteristic of those with current language difficulties, or whether it is also associated with past history of language delay.

Figure 2 shows relevant data. This shows that atypical lateralization is also seen in late bloomers, i.e., those who had been identified as late-talkers at 20 months of age, but subsequently improved. Four of these ten children had right hemisphere language and three had bilateral language. An unexpected finding was obtained when we subdivided the 'language difficulties' 4-year-olds into those who did or did not have earlier language delay. For those who had not been late talkers, referred to here as cases of language 'plateau', all six had right hemisphere ($N = 4$) or bilateral ($N = 2$) language. Language-impaired 4-year-olds who had been late talkers had predominant left-lateralization of language (3 left and 2 bilateral). However, the numbers are very small and it is possible that this was a chance effect. Typically-developing children who had never had language difficulties included 26 left-lateralized, 6 bilateral and 4 right-lateralized cases). In general, the data are consistent with the view that in a group of children who have either early language delay and/or language impairment at 4 years there is no overall bias to left-sided language lateralization. Figure 2 also shows data on handedness, confirming the lack of close relationship between writing hand and LI on the story-description task.

## Discussion

### Development of cerebral lateralization

We found no evidence to support the notion that cerebral lateralization for language gradually develops from a state of bilaterality to consistent left-sided processing in early childhood. Bilateral language was no more common in typically-developing 4-year-olds than in older children, and overall there was no evidence for changes in degree of lateralization with age on this task. This result is consistent with findings from a previous study of language lateralization using fTCD with 16 children aged from 2 to 9 years (*Lohmann et al., 2005*). It would be premature to dismiss maturational changes in language lateralization on the basis of these results: evidence for development of laterality

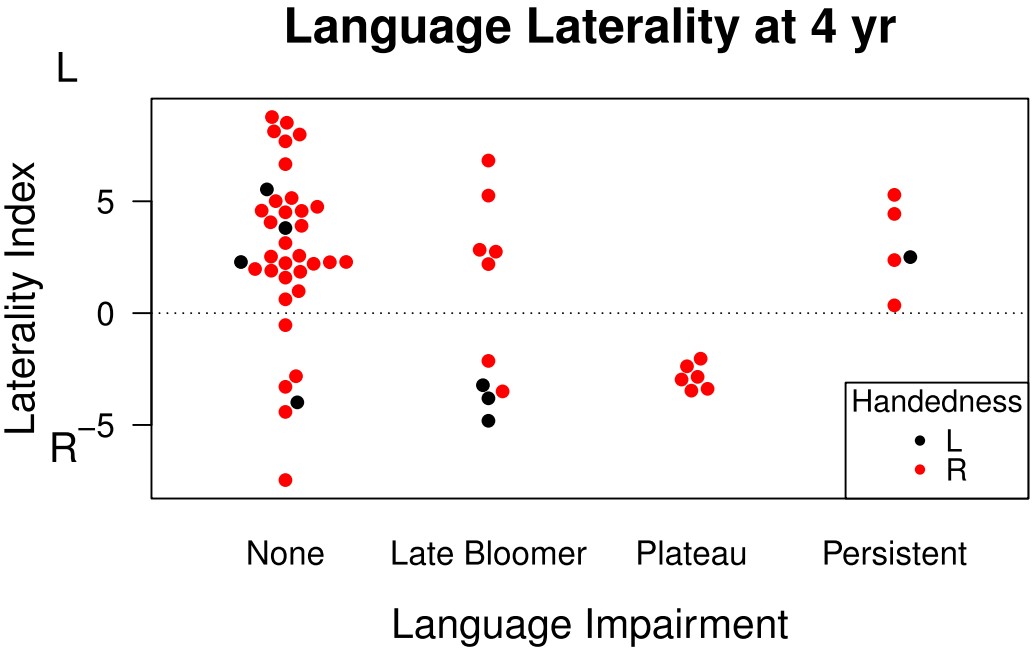

**Figure 2** Language laterality index on functional transcranial Doppler ultrasound by language status.

might be observed in a younger sample with more limited language skills, or by using different methods. Our result is discrepant with a previous study using fMRI in which the proportion of children with bilateral language decreased from age 4 to 12 years (*Berl et al., 2014*). Although low power of our study to detect weak associations could be an explanation for discrepant findings, it is also worth considering methodological differences between fMRI and fTCD. One advantage of our method is that the animation description task is naturalistic—describing a short episode that has just been viewed—and so may be less subject than word generation or semantic decision tasks to strategic effects, which could influence which brain regions are recruited for a task (*Berl, Vaidya & Gaillard, 2006*; *Brown et al., 2005*). Also, with fTCD one can define lateral bias in statistical terms, using the standard error of the LI across epochs in an individual, rather than adopting an arbitrary threshold. A major limitation of fTCD, however, is that it measures blood flow in the middle cerebral artery, which feeds extensive areas of cortex, and is not sensitive to regional variations in activation within a hemisphere. Studies using fMRI have indicated that lateralization of activation can vary from one language region to another (*Berl et al., 2014*; *Szaflarski et al., 2006a*). Furthermore, task-specific activation may become more focal with age (*Berl, Vaidya & Gaillard, 2006*; *Gaillard et al., 2000*). Such effects would not be detected using fTCD. On the other hand, fMRI studies may fail to detect activation outside a specific region of interest, and may be misleading if there is substantial variability from child to child in localization of activation (*Ahmad et al., 2003*).

Although we did not find an increase in left-sided lateralization over the age range from 4 to 12 years, it is possible that a decline in bilateral language would be observed if we extended observations into adolescence and adulthood. When an adult sample was tested

using the same task, only 3% had bilateral language and 15% right hemisphere language (*Bishop, Watt & Papadatou-Pastou, 2009*). However, note that much larger samples would be needed to give an adequately powered test of age trends, given that there is much wider variation in lateralization observed within any one age band than between age bands.

### Cerebral lateralization and language impairment

Turning to the findings on language-impaired children, our results showed that in the population at large, children with right-hemisphere or bilateral language lateralization are not at risk of language impairment, going against the notion that failure to establish left hemisphere language leads to language problems. On the other hand, we found that, at the group level, 4-year-olds with language impairment were not significantly lateralized for language. The number of children with language impairments was small, and it would be rash to place much weight on this finding if it had occurred in isolation. However, the consistency with previous studies using both fMRI and fTCD is noteworthy (*Badcock et al., 2011*; *Bernal & Altman, 2003*; *Chiron et al., 1999*; *Dawson et al., 1989*; *de Guibert et al., 2011*; *Ors et al., 2005*; *Tzourio et al., 1994*; *Whitehouse & Bishop, 2008*). It will be of interest to see how far these results using functional lateralization relate to structural asymmetries. Previous studies have reported rather poor agreement between structural and functional indices of brain asymmetry in unimpaired adult samples (*Dorsaint-Pierre et al., 2006*; *Keller et al., 2011*; *Propper et al., 2010*). However, it is possible that a closer relationship might be found in clinical samples, given that several studies have found reduced or reversed morphological brain asymmetry associated with developmental language impairments (*De Fossé et al., 2004*; *Gauger, Lombardino & Leonard, 1997*; *Herbert et al., 2005*; *Jernigan et al., 1991*; *Leonard et al., 2002*; *Plante et al., 1991*), though cf. *Preis et al. (1998)*.

Our results provide further evidence of the paradox noted by *Bishop (2013)*. Atypical cerebral lateralization is not associated with language impairment, but language impairment is associated with atypical cerebral lateralization. To make sense of this, we need to draw a clear distinction between the bias to left-sided language as it is manifest in individuals and in populations. The results obtained here contradict the idea that we can treat lack of lateralization in an individual person as a biological risk factor for language impairment. Insofar as lack of lateralization is associated with language impairment, it is at the level of the population, rather than the individual.

The Left Brain Bias model shown in Fig. 3 suggests that there is an etiological factor that puts children at risk for language impairment, and that at the same time removes the bias toward left hemisphere language that is found in most people. Importantly, there is no causal path from cerebral lateralization to language impairment: instead both are consequences of a common cause. A key feature of this model is its probabilistic nature. In a low-risk population, there is a factor that biases to left-sided language but this bias is not absolute—the establishing of lateralized language centres in the brain is illustrated in Fig. 3 as analogous to a process whereby balls drop through a hole into one of two containers; in low-risk cases, the hole is located so that around 90 per cent of balls drop into the left-hand container. In the high-risk group, however, there is no bias, so 50 per cent of balls drop into left and right containers.

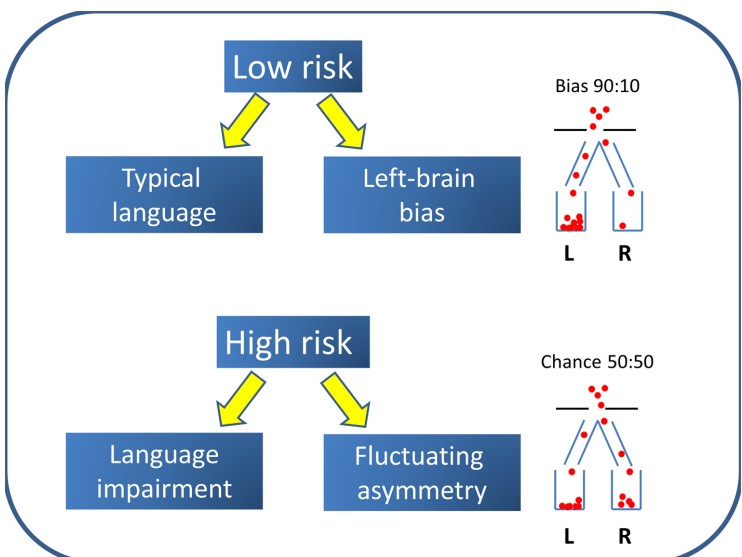

**Figure 3 Left Brain Bias model.** The population is a mixture of two subtypes: the majority have Left Brain Bias and a low risk of language impairment, whereas a high-risk minority have no bias to left-sided language and are likely to have difficulties with language learning.

The question then arises as to what removes the usual bias. One possible contender for a cause of this kind is brain injury. We know that early damage to left hemisphere language regions can lead to re-organization of language functions in response to pathology (*Rasmussen & Milner, 1977*; *Tillema et al., 2008*). This version of the Left Brain Bias model does not seem plausible, however, to explain common developmental language impairments, where there is little evidence of increased pre- or perinatal hazards (*Bishop, 1997*; *Tomblin, Smith & Zhang, 1997*), the majority of children have no hard neurological signs (*Tuchman, Rapin & Shinnar, 1991*) or evidence of pathology on brain scans (*Jernigan et al., 1991*; *Webster et al., 2008*), and twin studies suggest that genes play a major role in the etiology (*Bishop, 2006*). Note too that we had recruited only infants with no perinatal problems.

An alternative possibility is that genotypes associated with language impairment also affect cerebral lateralization, perhaps by affecting the timing of early stages of neurodevelopment. Most work on genetics of lateralization has focused on handedness rather than cerebral lateralization for language, because it is much easier to measure the phenotype and hence to study large samples. Single locus explanations of handedness are contradicted by results from genome-wide association studies (*Armour, Davison & McManus, 2014*), but this does not rule out a role for a range of genetic variants that could disrupt normal processes of lateralization. *McManus, Davison & Armour (2013)* proposed that rare mutations could disrupt the usual process that biases human laterality, so that instead there is fluctuating asymmetry—i.e., no bias to either left or right. Such mutations could also affect other aspects of neurodevelopment, including language acquisition. In this case, the cause of the link between handedness and language impairment would be rare genetic variants that had pleiotropic effects on brain development. Because different genes would

be involved in different people, the relevant genes would be hard to track down, but we might expect them to be involved in a common set of neural systems, cf. *Pinto et al. (2014)*.

This account has much in common with Annett's Right Shift Theory (*Annett, 1985*), a genetic model that was proposed to account for individual variation in handedness and cerebral lateralization. The key insight in Annett's model was that handedness phenotypes were best conceived of as biased to the right or unbiased, rather than left or right. However, the Left Brain Bias model differs from the Right Shift Theory in some critical respects:

(1) First, the Right Shift Theory treats handedness as an indirect indicator of cerebral lateralization. Annett maintained that handedness depended on a Right Shift gene, and the majority of the population had one or two copies of an allele that created a probabilistic bias both to left hemisphere language and to right-handedness. Those homozygous for the minor allele have no bias to left or right, in which case handedness and language laterality could be dissociated. Annett proposed that a measure of relative skill of the two hands could be used as an indirect indicator of genotype, and that different cognitive strengths and weaknesses were related to genotype. However, measures of relative hand skill do not differentiate language-typical from language-atypical children and do not show significant heritability in twins (*Bishop, 2001*; *Bishop, 2005*). Furthermore, relative hand skill on a peg-moving task is a poor indicator of language laterality as measured by fTCD (*Groen et al., 2013*).

(2) The Right Shift theory does not clearly distinguish between bias as it applies to a population and to an individual. In particular, when discussing handedness, Annett argues for a normal distribution of hemispheric bias, centred on zero for those who lack the 'right shift' allele, and shifted to the right for the rest of the population. Thus, an individual who lacks the right shift factor will not only be more likely to be left-handed, but also have, on average, more equal skill of the two hands. The Left Brain Bias model, in contrast, applies at the level of the population, not to the individual. Thus people who lack Left Brain Bias do not necessarily have more symmetric language representation; their cerebral asymmetry may be equal in magnitude to that of those with Left Brain Bias, but the direction of that asymmetry is equally likely to be left or right, rather than predominantly left.

(3) When discussing cognitive effects of cerebral lateralization, the Right Shift Theory is not always clear about whether postulated language deficits are consequences of atypical cerebral lateralization itself, or of the genes that influence cerebral lateralization. For instance, *Annett (1998)* stated that "The question is whether cognitive, social, and motor skills vary with different patterns of cerebral specialization", yet she also noted that whatever leads to unbiased cerebral dominance (RS−− genotypes) leads to risk for speech problems and dyslexia. This is a subtle distinction, but it is crucial for making sense of observed data. Although evolutionary considerations might lead one to expect that left-sided language lateralization might be advantageous, evidence for this notion from non-clinical populations is at best weak and inconsistent (*Groen et al., 2012*; *Knecht et al., 2001*; *Powell, Kemp & García-Finaña, 2012*). The Left

Brain Bias model proposes that atypical cerebral lateralization is not itself a causal factor in language problems. Thus this model does not have an arrow showing a direct link from language lateralization to language impairment. Rather than seeing atypical cerebral lateralization as an 'endophenotype' for language impairment (*Bishop, 2013*), it is a marker of a risk factor that is associated with language problems. But it is a very imprecise marker, because it works by altering the proportions with left and right hemisphere language in those with the risk factor. If it does not affect the degree of asymmetry in individuals, then 50 per cent of those with the risk factor will have typical left-hemisphere language.

A corollary of these propositions is that, if there is a genetic influence on Left Brain Bias, this will be difficult to demonstrate using conventional approaches. *Bishop (2013)* argued that the extent of genetic influence on cerebral lateralization may have been overestimated, given that neither genome-wide association studies nor twin studies had found evidence for strong genetic effects on measures of structural or functional asymmetry: see also *Eyler et al. (2014)*. This dismissal may, however, be premature; the difficulty of demonstrating a genetic influence may lie in the probabilistic nature of the phenotype. If a child lacks any factor biasing to left or right, then chance will determine lateralization, and this is equally likely to yield discordant as concordant twin pairs. A classic twin analysis might then lead to the conclusion that non-shared environmental influences are the main determinant of an individual's brain asymmetry, if there are many discordant monozygotic twin pairs. However, to test a genetic version of the Left Brain Bias model, we would need a different measure of the phenotype—one that more directly indicated whether the person came from a population with a 50:50 bias or a 90:10 bias.

Our results suggest several new research approaches. First, we are now in a position to conduct studies of children using direct assessment of cerebral lateralization, and there is evidence that language laterality can be measured reliably, even in pre-schoolers. It would be informative to conduct such studies with even younger children, but our experience to date has found this to be challenging. Toddlers will often not tolerate the ultrasound headset, and will not speak and remain silent on command. New paradigms need to be developed to overcome these limitations. Second, as argued by *Bishop (2013)*, it is likely that cerebral lateralization is not a unitary phenomenon. Even within the domain of language, the same person may show different directions of lateralization (*Annett & Alexander, 1996*). This is often thought to just reflect measurement error, but it may be meaningful, and by studying both structural and functional aspects of lateralization, we may be able to identify laterality profiles that can help identify presence or absence of Left Brain Bias at the individual level. This would be an important advance, as currently we do not have a useful phenotype for studying genetic origins of individual differences. Finally, if Left Brain Bias is a pleiotropic consequence of a range of rare genetic variants that increase risk of language impairment, then this suggests that it may be fruitful to look at genes that influence body plan asymmetry when seeking genetic variants that are associated with language disorders (*Brandler & Paracchini, 2014*).

## CONCLUSIONS

This study has confirmed that there is an unusual pattern of association between functional cerebral lateralization and language skills: children with atypical cerebral lateralization do not differ significantly from other children in language abilities, but children with developmental language impairments lack the usual bias to left-sided language representation at the population level. These results fit with a model that maintains there is no cognitive disadvantage caused by atypical lateralization per se, but rather that there is an etiological factor that simultaneously increases the risk of language impairment and reduces the usual population bias to left hemisphere language.

## ACKNOWLEDGEMENTS

This study would not have been possible without the generosity of the families who took part. We also thank Helen Watt, Elizabeth Line, David McDonald and Sarah McDonald who recruited and assessed the sample of toddlers.

### Funding

This work was supported by Programme Grant no. 082498/Z/07/Z from the Wellcome Trust. The funders had no role in study design, data collection and analysis, decision to publish, or preparation of the manuscript.

### Grant Disclosures

The following grant information was disclosed by the authors:
Wellcome Trust: 082498/Z/07/Z.

### Competing Interests

The authors declare there are no competing interests.

### Author Contributions

- Dorothy V.M. Bishop conceived of the study, analysed the data, wrote the paper, developed the child-friendly procedure using fTCD and read and commented upon the draft manuscript.
- Georgina Holt developed the child-friendly procedure using fTCD, tested children at 4 years of age and assisted with data analysis, and read and commented upon the draft manuscript.
- Andrew J.O. Whitehouse and Margriet Groen collected data with older children, and read and commented upon the draft manuscript.

### Human Ethics

The following information was supplied relating to ethical approvals (i.e., approving body and any reference numbers):

This project was approved by Oxfordshire Research Ethics Committee A, file number A03.025.

## Data Availability

The following information was supplied regarding the deposition of related data:
Open Science framework: https://osf.io/yv4ra/.

## Supplemental Information

Supplemental information for this article can be found online at http://dx.doi.org/10.7717/peerj.507#supplemental-information.

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
