# Peer review of "No population bias to left-hemisphere language in 4-year-olds with language impairment"

_PeerJ, doi:10.7717/peerj.507_

## Round 0.1 · original submission · Minor Revisions

I carefully read the manuscript together with prior reviews and the authors' replies. I think the authors effectively managed to resolve all the issues raised by the first round of reviews. The manuscript is now almost ready for publication.

Besides noticing some minor language slips (e.g., "an small but significant", row 109), which just need a careful re-reading, I would suggest the authors to try and better characterise the flow of information in the Results section. In particular, the passage between the results pertaining to Question 1 and those relating to Questions 2 and 3 should be highlighted, perhaps resorting to the tags in Figure 1 for the latter. This way, it would become clearer early on that in answering Question 3 the authors pulled together the Language Plateau, Late Bloomers and Persistent SLI subgroups (now this is reminded to the reader only at rows 391-2). Also, in the abstract the authors conclude with a statement "cognitive impairment". I think they meant "language impairment".

As for the second row of reviews, I suggest the following:
- Reviewer 1: Please discuss and cite Szflarski et al., 2012, if deemed relevant, and fill in remaining "sweeping statements with no literature backup". I agree with the authors that it is not necessary to introduce the literature on structural data on language lateralisation.
- Reviewer 2: The figure deposited in the LabArchives constitutes Supplemental Information (please expunge from References). The issue about Crow's paper is indeed debatable, but I am inclined to agree with the authors' decision of not including that particular paper.

External reviews were received for this submission. These reviews were used by the Editor when they made their decision, and can be downloaded below.

---

## Round 0.2 · accepted · Accept

I am satisfied with the final changes and deem the paper suitable for publication in PeerJ.